# Tannin in Ruminant Nutrition: Review

**DOI:** 10.3390/molecules27238273

**Published:** 2022-11-27

**Authors:** Maghsoud Besharati, Aristide Maggiolino, Valiollah Palangi, Adem Kaya, Muhammad Jabbar, Hüseyin Eseceli, Pasquale De Palo, Jose M. Lorenzo

**Affiliations:** 1Department of Animal Science, Ahar Faculty of Agriculture and Natural Resources, University of Tabriz, Ahar 5451785354, Iran; 2Department of Veterinary Medicine, University of Bari A. Moro, 70010 Valenzano, Italy; 3Department of Animal Science, Agricultural Faculty, Ataturk University, Erzurum 25240, Turkey; 4Department of Zoology, Faculty of Biosciences, Cholistan University of Veterinary and Animal Sciences, Bahawalpur 63100, Pakistan; 5Department of Nutrition Sciences, Faculty of Health Sciences, Bandirma Onyedi Eylul University, Balikesir 10200, Turkey; 6Centro Tecnológico de la Carne de Galicia, Avd. Galicia 4, Parque Tecnológico de Galicia, 32900 Ourense, Spain; 7Área de Tecnología de los Alimentos, Facultad de Ciencias de Ourense, Universidade de Vigo, 32004 Ourense, Spain

**Keywords:** tannins, ruminants

## Abstract

Tannins are polyphenols characterized by different molecular weights that plants are able to synthetize during their secondary metabolism. Macromolecules (proteins, structural carbohydrates and starch) can link tannins and their digestion can decrease. Tannins can be classified into two groups: hydrolysable tannins and condensed tannins. Tannins are polyphenols, which can directly or indirectly affect intake and digestion. Their ability to bind molecules and form complexes depends on the structure of polyphenols and on the macromolecule involved. Tannins have long been known to be an “anti-nutritional agent” in monogastric and poultry animals. Using good tannins’ proper application protocols helped the researchers observe positive effects on the intestinal microbial ecosystem, gut health, and animal production. Plant tannins are used as an alternative to in-feed antibiotics, and many factors have been described by researchers which contribute to the variability in their efficiencies. The objective of this study was to review the literature about tannins, their effects and use in ruminant nutrition.

## 1. Introduction

Tannins are polyphenolic compounds and the secondary metabolites of higher plants [1,2]. They are found naturally in the structures of plants and their molecular weight can vary between 300 and 5000 daltons. Higher-plant secondary metabolites have a significant polyphenolic structure which does not justify the specifications for placement in the tannin classification. Plant tannins have a phenolic effect (e.g., blue color, iron (III) chloride) and as a result of water-soluble polyphenols with molarities ranging from 300 to 3000, catalyzing alkaloids, gelatins, and other proteins [3]. Tannins are a diverse group of secondary plant metabolites that can be dissolved in polar solutions and are differentiated from polyphenols by their ability to dissolve in polarized solutions [4]. These compounds appear to have no function in plant metabolisms such as biosynthesis and energy exchange. Still, they are responsible for various biological activities. Tannins can prevent ruminal nitrogen metabolism by decreasing proteolytic bacteria [5,6]. Moreover, they can reduce the ruminal bio-hydrogenation process and increase the flow of unsaturated fatty acids to the duodenum [7]. Additionally, phenolic compounds and tannins, through selective activity on ruminal bacteria, can alter the process of ruminal bio-hydrogenation and increase the amount of conjugated linoleic acid in meat and milk production [8], but this effect is dosage-dependent. As a result, tannins improve the efficiency of nitrogen use by retaining more nitrogen in the body [6,9,10].

Ruminants, both intended for meat and milk production, are of significant economic interest in the world, with different production systems. Moreover, their production has been based on tannin-rich feed (Table 1) since ancient times, and actually, whether an extensive or intensive technique is applied, forages rich in tannins (Table 2) or rations with the addition of vegetable co-products rich in tannins (Table 3) are used in order to obtain positive effects on production, quality and animal welfare [11]. As it is clear, the potential sources of tannins for ruminants are plentiful.

Several factors can modulate tannins effect as negative, positive or indifferent. This can be due to their different chemical composition or type, the administered quantity, the animal species involved and the diet to which tannins are added [73]. For these reasons, there are many aspects that could have affected the apparently inconsistent findings that can be found in literature. They are characterized by great chemical structural diversity, with results that cannot be easily extrapolated from one type of tannin to others [74].

Tannins may have important effects, both adverse and beneficial, on animal performance and on their production quality, due to chemical structure but also to their content in the diet; both aspects are intrinsically linked to animal physiology [75,76]. Some adverse effects may be listed as a reduction of feed intake, fiber digestibility and consequently animal performance [77], but some other researchers showed how tannins may enhance protein utilization, control internal parasites and act also on production (growth performance, milk, meat, wool) [75,78] and animal welfare [79,80]. Moreover, it has been shown that they may play an important role in improving antioxidant and immunity status of animals [57,81].

Recent studies have seen an increasing interest in this topic, minding that they can be used in nutritional strategies, but limits on their practical application are linked to potential adverse effects.

With a focus on ruminants, the main objective of this paper is to review the current knowledge about dietary tannins use and effects on animal production and performance (both milk and meat). Of course, tannins’ chemistry and their chemical properties will be discussed.

## 2. Classification of the Tannins

The tannins have several structures that can provide a systematic classification system with their properties and chemical structural basis for further analysis [82,83]. Many tannins can be hydrolyzed, and these are classified as “hydrolyzable tannins” (HT). Polymeric proanthocyanidins and non-hydrolyzable oligomeric were classified as condensed tannins [84,85]. The ‘HT’ included both the gallotannins and the ellagitannins [86,87]. Due to a different C–C interaction for polyphenolic waste and polyol, ellagitannins may not be hydrolyzed but classified as HT for specific purposes. In addition to hexahydroxydiphenoyl (HHDP), the first tannins were described in 1985 as having the C-glycosidic catechin units (the structural and functional component of isomeric HT). Because the only component of the C–C relationship between their catechin and glycosidic entities is partially hydrolyzable, these tannins were previously categorized as “non-classified.” The terms ‘complex tannins’ and flavanoellagitannins have been established over the following years to properly merge these ‘non-classified tannins’ [88,89]. Hydrolyzable and non-hydrolyzable or condensed tannins are the two types of tannins that can be found. However, they are not structurally diverse [90,91]. Tannins have catechin units flavanotannins or condensed flavonoids that characterize the tannins’ [92,93]. Paired flavan-3-ol (catechin) units are used in condensed tannins—polymer flavanotannins (oligomeric and polymeric proanthocyanidins). On a structural level, tannins are split into four categories: gallotannins, ellagitannins, complicated tannins, and condensed tannins (Figure 1) [94,95].
(1)Gallotannins tannins are derivatives of diverse polyol, catechin, or triterpenoid units of galloyl or their meta-depends [98].(2)Ellagitannins tannins are two C-C galloyls and have no glycosidically-related catechin unit attached to one another [99].(3)Complex tannins are bound glycosidically by a catechin component of gallotannine or ellagitannin [100].(4)Condensed tannins are all proanthocyanidins oligomeric and polymeric formed by the similarity between C-4 and one C-8 or C-6 of one catechin and the next catechin monomeric [101].

## 3. Chemical Properties of Tannins

Tannins are polyphenolic compounds also known as tannic acids. It is the name given to amorphous (amorphous) substances in the form of a light yellow-brown powder, flake or spongy mass obtained from plants. Tannins are solid compounds in phenol structure and are soluble in water. They are usually found in the roots, wood, bark, leaves and fruits of plants [102,103].

Tannins are chemically divided into hydrolyzable tannins (HT) and condensed tannins (CT). The HT yield water-soluble compounds such as gallic acid, pyrocatechic acid and sugar as hydrolysis in the presence of an acid or enzyme. It is slightly soluble in water, well soluble in alcohol and acetone. One of the best-known examples of HT is gallotanins. The CT, which are a much larger group, cannot be hydrolyzed. They form dark red insoluble compounds called *flobafen* with strong acids or some oxidizing agents against heat [102].

Hydrolyzable tannins contain hydroxyl groups esterified with carbohydrate (usually D glucose) and phenolic groups in the center. They decompose into carbohydrates and phenolic acids as a result of hydrolysis by weak acids, weak bases, hot water or enzymes such as tannase [104]. They are usually found in fruit seeds, in low amounts. They are degraded by the enzymatic functions of the ruminal microflora and by gastric digestion and turn into low molecular weight, absorbable toxic metabolites. As a result of degradation, gallic acid, pyrogallol, phloroglucinol and finally acetate and butyrate are formed as a result of successive enzyme activities [104,105].

Proanthocyanidins, also known as condensed tannins due to their chemical structure, are the most common tannin group found in trees and shrubs used as fodder plants [103]. These do not carry carbohydrates in the core; they are oligomers or polymers of carbon-carbon bonded flavonoid units (e.g., flavan-3-ol) resistant to degradation by hydrolysis [105]. Depending on the chemical structure and degree of polymerization of proanthocyanidins, their solubility in aqueous organic solvents varies. The term proanthocyanidin derives from the acid that catalyzes the oxidation reaction that results in the formation of red anthocyanidins by heating proanthocyanidins in acidic alcohol solutions. The most well-known anthocyanidins produced are cyanidin and delphinidin. Catechic tannins, which are not hydrolyzed by the action of enzymes or dilute acids, are the condensation product of catechin and turn into pyrocatechol by dry distillation [106].

The high affinity of protein tannins depends on how many phenolic compounds bind many places with carbonyl peptide groups [107,108,109].

The formation of these complexes is specific to the level of affinity between the participating molecules, which are based on each tannin’s chemical characteristics and the protein concerned [110,111,112]. For tannins, their relatively high molecular weight and chemical structure flexibility are some of the factors that support complexes’ development [113,114,115]. In addition, it was reported that the addition of condensed tannin (CT) extract to a forage-based diet can contribute to the decrease in CH_4_ emission by ruminants [10,82,116,117]. However, many types of condensed tannin with different biological activities, such as CH_4_ emissions, may impact animal responses [115]. The importance of the condensed tannin source and the dietary level must be understood for a better understanding of CH_4_ emission by ruminants and for the implementation of effective feeder methods that take benefit of such implications.

The small-statured and hydrophobic proteins have an open, flexible structure and are rich in proline [118,119,120]. Generally, tannins and proteins have very unstable bonds, which break and re-form continually. Kumar and Singh [121] showed that the formation of these structures might well involve four types of bonds.

The hydrogen linkages (reversible and pH-dependent) between phenolic hydroxyl radicals and amide oxygen in peptide-protein bonds are the same. The aromatic ring of the phenols interacts hydrophobically with hydrophobic regions of the protein (reversible and dependent on pH). By forming ionic (reversible) bonds between the phenolic ion and the cationic region of the protein (exclusive to HT). Covalent (irreversible) connection to quinones with oxidation and subsequent condensation of polyphenols with nucleophile protein groups. Hydrogen bonds were involved in forming tannin protein complexes for a long period, but later research demonstrated the significance of hydrophobic interactions.

## 4. Effect of Tannins on Ruminant Nutrition

It is important to monitor the feed consumption, the structure and molecular weight of the compound, and the physiology of consuming species to understand the beneficial and damaging impact of tannins on ruminants [116,122]. Distinct analytical methods, especially different standards, may yield different results (e.g., Equitrac, tannic acid, catechin, cyanidin, delphinidin, or the plant itself internal standards, etc.) [123]. Tannins are abundant in forages, trees, shrubs, and legumes often ingested by ruminants, particularly condensed tannins. As a result, the effects of condensed tannin on ruminant nutrition, health, and productivity have been extensively researched [73,115,124]. The beneficial or adverse effects of condensed tannins, particularly the concentration of crude protein (CP) in the diet, on ruminants are based on their amount, type, chemical structure, and feed composition [114] (Figure 2).

Frothy bloat is a common gastrointestinal disease in ruminants, which is caused by the accumulation of gas in the rumen and reticulum, causes damage to the function of the gastrointestinal tract and respiratory tract, which can be reduced by using tannins [125]. Li et al. [126] reported that a bloat of 1.0 mg CT/g of dry matter (DM) efficiently prevents pasture. Alfalfa pasture bloat can be controlled by incorporating condensed tannin-containing fodder, such as sainfoin, into the crop. Tannins have also been reported to help ruminants control parasites in their digestive tracts [127]. Hoste et al. [128] have reported in vivo studies on significant anthemia effects in the gastrointestinal tract of sheep, goats and deer on tannins in sainfoin, sulla, *Acacia nilotica*, *Sericea lespedeza*, *L. pedunculatus,* and chicory.

### 4.1. Effect of Tannins on Voluntary Feed Intake

The researchers hypothesized that tannin ingestion would limit ruminant voluntary feed intake. As a result, researchers now have a lot more information and can make more accurate conclusions about tannins, their dosages, and their effects on the animals which consume them [129,130].

The voluntary feed intake of ruminants by providing plants with high condensed tannin (CT) content (usually >50 g/kg DM) can be reduced, with a moderate or low intake (<50g/kg DM) not affecting it [131]. On this side, Cabral Filho et al. [132] observed no effect on DM consumption of sheep.

Voluntary feed consumption in ruminants depends on the taste and degree of digestion of the feed. With the consumption of tannin-containing feeds, tannin forms a compound with the glycoprotein found in the saliva. A high amount of tannin reduces the flavor and consumption of the feed. Therefore, it causes a decrease in productivity in animals. Some animals can adapt to diets containing high amounts of tannin by increasing the quantity of proline-rich proteins in the saliva. These proline-rich proteins in saliva form bonds with tannin. Thus, it prevents tannin from making compounds with proteins in the diet [133]. Barry and McNabb [134] reported a detrimental effect on voluntary feeding in grazing sheep given *Lotus pedunculatus* (condensed tannin content > 50 g/kg DM), but no findings in *L. corniculatus* (condensed tannin content 34–44 g/kg DM)

The consumption of tannin-containing feeds for a long time causes enlargement of the salivary gland [135]. Not all animals can secrete proline-rich proteins in saliva. For example, animals such as humans, mice, deer, and gazelles secrete proline-rich protein in their saliva. However, this secretion is much less in sheep and cattle [136]. It has been reported by several researchers that feed consumption decreased in sheep fed diets containing high amounts of tannin. Waghorn, Shelton, McNabb and McCutcheon [131] reported that feed consumption decreased in sheep fed with feed containing 55 g of tannin per kilogram. It has been reported that the amount of tannin below this level has little or no effect on the feed consumption of the animal.

The effect of condensed tannin varies according to the ruminant species. They do not contain proline-rich protein in the saliva of domesticated sheep and cattle [136]. For this reason, it can be said that domesticated sheep and cattle may be more affected by tannin.

Hydrolyzable tannins have also had variable effects depending upon the consumed quantity. McSweeney et al. [137] observed that *Terminalia oblongata* having low HT (34 g kg^-1^ DM), when supplemented to sheep, shows no significant reduction in voluntary feed intake (34 g kg^−1^ DM) while *Clidemia hirta*, a shrub which is high with HT (>50 g kg^−1^ DM) shows a reduction in voluntary feed action in the same animal. According to Frutos et al. [138], voluntary food intake of soya bean treated sheep with HT (20.8 g HT/kg DM diet) did not decrease. However, in an experiment with sheep fed 8 g Tannic Acid per kg live weight, voluntary feed consumption fell considerably within 24 h (from 18 to 2.5 g DM/kg LW).

Feed palatability, sluggish digesting, and enhanced conditioned aversions are three recommendations for clarifying the harmful effects of excessive tannin concentrations on voluntary feed intake [138].

The reaction between salivary tannins and mucoproteins and the direct response to taste receptors can reduce the taste, an astringent sensation [121]. The saliva of many herbivore species consists of proline-rich tannins and proteins that are plant species components and can bind with tannins [139]. The tannin-proline-rich protein complexes, which are stable over the whole pH range of the digestive system, are responsible for building stable complexes other than protein-tannins, which have a negative influence on taste and feeding and promote digestion of tannin-rich diets [140].

Herbivores are thought to have evolved distinct adaptation mechanisms for tannin-rich plants through time [140,141]. All the time animals secrete proline-rich proteins, but sheep only produce them when they consume tannin-rich plants [136]. However, in cows, tannin consumption did not result in increased synthesis of these proteins, despite the presence of many proteins with high partiality in their saliva [142].

Narjisse, Elhonsali and Olsen [140] directly transplanted tannins into the rumen to test if non-rational causes were to blame for the voluntary feeding intake reduction. The digestive system’s discharge is harmed when dry matter digestion is slowed. This sends out signals that the animal is “overflowing” and gives the nerve center feedback. Researchers have determined that this might impact voluntary feed intake in ways other than diminished palatability [131].

Thirdly, the detrimental post-prandial effects of tannin consumption and the establishment of conditioned aversions are found [143]. Rumen microbiota plays a fundamental role in ruminant nutrition, and factors related to microbial fermentation are assumed to mediate the post-prandial consequences of ingestion of tannin-rich feeder [138].

### 4.2. Effect of Tannins on Digestibility of the Diet

There are many articles on tannins’ ability to reduce food digestibility. Tannins mainly affect proteins (Figure 3). Tannins were also found to make slight changes to different degrees in other components [121]. The effect of tannins on protein is based on hydrogen bonds that are stable between pH 3.5 and 8. Ruminal pH changes affect the complexes and provide information about the function of tannins in the gastrointestinal tract [103]. Cabral Filho et al. [132] investigated three levels of tannin in the diet of sheep and reported a different DM digestibility between high- and low-tannin cultivars. The low tannin content diet had a higher crude protein digestibility, and there were no significant differences between the medium tannin content and high tannin content diets. The highest values for low tannin content and high tannin content were found for the neutral detergent fiber (NDF) digestibility, and the average medium tannin content diet was 582 g/kg, which differed only a little from the low tannin content diet.

Tannin ingestion modifies digestibility by the correlation of the ruminal fermentation pattern. These effects are addressed in two sub-sections, but the repeated conclusion is that the higher dietary tenor of tannin reduces food digestibility by increasing fecal nitrogen excretion. In any case, increased release of endogenous proteins such as salivary glycoproteins, body fluids, and gastric catalysts, and increased descaling of intestinal cells, are detrimental consequences of tannin consumption [131]. This rise in fecal nitrogen might be due to an increase in metabolic fecal nitrogen, i.e., endogenous nitrogen that does not account for a decrease in the amount of protein ingested from the feed.

According to Besharati and Taghizadeh [144], increased dried grape by-products in diets shows a negative effect on CP digestibility. It was assumed that tannins could influence feed protein and rumen microbiota; the accessibility of rumen microbiota to such proteins has decreased because tannins bind with protein. The same authors [144] analyzed four diets, one as a basal diet (alfalfa) and the remainder as mixed diets (dried grape by-product with alfalfa). Table 4 shows the digestibility of the chemical components of the in vivo study. Increased products of dried grape in diets lead to lower digestibility, but no difference was found. Incremental grapes substituted by alfalfa levels resulted in the digestibility of DM by quadratic and cubic, but no linear effect was observed. Dry grape by-product supplementation has no impact on organic matter (OM) digestibility in experimental diets. An increase in dried grape by-products was associated with no significant response, and a quadratic response was found. In addition, the digestibility of CP was affected by dried grape by-product in basal diets, as well as an increase in the levels of supplementation of dried grapes by-product in CP digestibility in diet. There was no indication of a quadratic or cubic impact. In treatments, the treated and control group four had the greatest and lowest digestible CP, respectively. The addition of dried grape by-products to treatments reduces the digestibility of NDF and acid detergent fiber (ADF) with increased dried grape by-products to foods. A linear increase in the amount of additional dried by-products affected the digestibility of ADF and NDF in diets.

McSweeney et al. [137] investigated the effects of increasing condensed tannin in the diet in sheep, finding that increasing condensed tannin in the diet (from 6 to 65 g/kg DM) lowered N digestibility from 0.805 to 0.378 and raised excretory N in sheep feces from 4.3 to 9.7 g/d. Hagerman et al. [103] showed that tannins could reduce CP digestibility. The significant drop in N digestibility was reported to be due to the tannin presence, and the results were similar to those in *Lotus pedunculatus*-fed sheep and the ryegrass in *Lotus pedunculatus* with or without polyethanol glycol (PEG) [131].

If the drop in N digestibility was attributable to a decrease in protein digestion, the decreased protein digestibility might account for the overall reduction in tannin-related DM digestibility [144].

Yisehak et al. [145] demonstrated that PEG addition in the diet as a tannin binder enhanced digestion and performance in both species but with the largest effect size in sheep. Corrêa et al. [146] showed that tannins are useful to reduce intakes of feed but to increase daily feed time linearly, even though the number of meals did not change. Tannins also decreased DM, CP, OM, urinary urea, and TDN digestibility linearly, while reducing ADF and NDF was quadratic. Tannins reduced rumen loss by slowing transit and digesting rates in a linear fashion. They concluded that tannins might have a significant effect in lowering nitrogen excretion in urine.

### 4.3. Effect of Tannins on Ruminal Microbiome and Fermentation

The rumen microbiome had great importance, because through rumination, fibrous and non-fibrous plant material can be transformed into valuable products such as meat and milk, thanks to the rumen microbiome activity [147]. Rumen inhabits various microbes like bacteria, protozoa, fungi, archaea, and bacteriophages [148]. The rumen microbiome and host had symbiotic relationships, each one provide substances to the other: first of all, mastication and rumination are processes that permit to expand feed surface area to microbial attachment, mixing constantly digesta and yielding fermentation products as volatile fatty acids compounds (VFA); moreover, rumen permits the elimination of useful and toxic substances maintaining the best condition for microbial growth and activity, returning urea for microbial development (Figure 4) [147].

Rumen microbiome is mainly made up of bacteria, with a population that exceeds one thousand cells per gram of rumen content and includes over 200 species [148]. Their main role is the metabolism of carbohydrates and nitrogen taken with the diet. Methane emission is mainly dependent on H2-producing bacteria [149].

Fungi represents a range of 5 to 20% of the total ruminal microbiota [150,151]. The anaerobic ones play key role in lignocellulosic fiber breakdown [152], considered one of the most able fiber-degrading agents [153], they produces a lot of enzymes (cellulases, xylanases, mannanases, esterases, glucosidases, and glucanases) that are important for plant polymers breakdown; but they are also characterized by amylolytic [154] and proteolytic activities [155].

Ruminal protozoa, commonly divided in flagellates and ciliates, represent about 20–50% of the total rumen microbiome [156]. Instead, that of ciliates is particularly important because it engulfs carbohydrates and do not allow abnormal rumen fermentations which could alter the pH and make the rumen environment uncomfortable for the microbiota [157]. They have a positive correlation with CH4 production because they are involved in H2 production that is subsequently converted to CH4 by the methanogens through the hydrogenotrophic pathway [158], so their defaunation is able to reduce CH4 emission [157,159]. Archea, the third rumen domain, constitutes about the 20% of rumen microbiome [160]. They are involved in methanogenesis [160,161], starting by different substrates as formate, or acetate, methanol, H2, methylamines, and CO2 (71).

In a large pH range (3.5-7), tannins can bind dietary proteins, starch and sugars creating strong complexes [162], and this can limit their digestibility if concentration is more than 5% [163]. Although many studies have been conducted, the real action of tannins in rumen is not yet clear [116]. In fact, although they possess bacteriostatic effects, their association with rumen microbiota is different due to different tannins nature. In fact, hydrolysable tannins resulted more hydrolysable by microbiome than condensed ones [137]. Tannins can inhibit methanogens activity directly, limiting the degree of microbial hydrolysis, and indirectly decreasing the H2 availability [164], although this can reduce fiber digestion. Their ability to modify the ruminal microbiome can result in a reduction in protein degradation, can reduce methanogenesis and inhibit the fatty acids biohydrogenation [165,166,167] (Figure 5).

As expected, results about potential shifting of microbial population and rumen fermentation are variable depending on species, doses, and tannins source. Researchers reported different results about methane mitigation. Some studies reported that tannins can reduce the short-chain fatty acids and acetate production, increasing the diversity and abundance of butyrate-producing bacteria and other beneficial species (such as probiotic species), and enhancing the amino acid metabolic pathways [146]. Although these exhibit different effects, the mechanism action is not known. Tannin supplementations seems to reduce the crude protein digestion and methane production in beef [168], although other authors reported no effect on live and growth performance of beef, no effect on in vitro digestion traits, and no reduction in CH4 emission [57]. Different tannins sources have been used with the aim to mitigate methane emission without affecting, but improving, productive performance. For example, gallic acid has the potential to reduce the rumen greenhouses production without affecting performances [169]. The same results were found for tara, mimosa, gambier extracts [170] and legumes with condensed tannins [171]. However, dietary sources of tannins and their chemical composition can affect quiet differently rumen microbiota, in fact HT are considered more suitable than CT for methane mitigation [164]. The encapsulation can be considered a good solution for tannin extract due to the slow release of tannin and their more efficient utilization [172,173]. However, CT rich diets can reduce CH4 emissions in ruminants, with an effective inhibitory effect on ruminal protein degradation and CH4 emission, but this requires careful source selection for diet inclusion to avoid adverse effects on feed digestibility and efficiency [174]. Nevertheless, many other studies are necessary in order to fully understand the mechanism of action of tannins at the rumen level regarding the potential inhibitory effect of methanogens and protozoa, and above all to determine the best inclusion level to inhibit the production of methane without compromising the productions. The tannins are characterized by a powerful antioxidant effect as they can scavenge free radicals due to hydroxyl groups, degree of polymerization, and redox activities [175,176]. In fact, tannin supplementation improves ruminant’s antioxidant state [57,177,178,179]. Among the chemical characteristics, hydrolysable tannins are considered the most potent antioxidants [180].

Tannins are the main source for the degradation of the ruminal protein [103]. Tannins are intimately connected to these molecules, and the pH of the ruminal media encourages tannin protein complex synthesis. Protein degradation can be reduced as a result of lower ammonia nitrogen synthesis and more non-ammonia flow to the duodenums [131].

Tannins influence proteins and carbohydrates, particularly hemicelluloses, celluloses, starch, and pectins [181]. The tannins have long been a secondary anti-nutritional effect on fiber degradation [137,182].

The methods for minimizing ruminal damage to various dietary tannin components are unknown. The most widely established procedures are denial of nutrients, enzyme inhibition, and direct action on rumen bacteria [183,184]. Regarding the first of these, various authors have indicated that attachment of rumen microbes to plant cell walls is necessary for the breakdown process, but the presence of tannins in plant material prevents the plant material from adhering to rumen microorganisms [185,186]. Furthermore, complexes of proteins and carbohydrates decrease the effectiveness with which these nutrients may reach these microbes [187]. Tannins also limit the availability of several metal ions that are required for rumen microbial metabolism [184].

Tannins also inhibit the enzymatic activity of rumen microbes [137,186]. But the exact mechanism of enzyme inhibition activity of tannins is still poorly understood. Polyphenolics have been found to react to the cell wall of bacteria and to secrete extracellular enzymes. Any interaction will probably inhibit the transport of nutrients into the cell and delay the organism’s growth. Although the exact mechanism of how tannins bind to enzymes, whether bacterial or endogenous, does not indicate inhibition, some writers have reported that tannins modify bacterial proteolytic, cellulolytic, and other enzyme activity [142,188]. When fibrolytic enzymes were investigated, condensed tannin was more effective in inhibiting hemicellulases than cellulases [143]. This is since they relate to bacterial cell walls, whereas hemicellulases are extracellular and more susceptible [135]. This explains why most investigators have reported a greater decrease in hemicellulose degradability in tannin presence [131,182]. However, this can vary depending on the tannin desired [186].

Besharati and Taghizadeh [144] indicated tannin contents in the plant material reduce fermentation, resulting in major differences among VFA and NH3–N concentrations. Priolo et al. [189] reported that when PEG and tannin-feed was given to sheep, rapid rumen fermentation was observed by PEG, resulting in significant differences among VFA and NH3–N concentrations.

CT/bacterial cell reactions are less well-conceived than CT/feed protein reactions. However, the interactions between condensed tannin and the substrate appear to be different between bacterial cells and Rubisco. The CT-bacterial interaction was found much stronger than CT-protein interaction, or the interaction is of a different kind [190]. The condensed tannin most likely connects the bacterial cell surface to limit the action of cell-bound enzymes. The inhibition scope may vary with different condensed tannin types. However, the fact that we cannot quantify actual condensed tannin contents of the CT-bacterial interactions requires more research since digesta affects our knowledge of physical and chemical relationships between CT, rumen microbes, and plant protein [191].

Cabral Filho et al. [132] observed no significant differences between the treatment groups with diets containing three different concentrations of tannin. Aboagye, Oba, Castillo, Koenig, Iwaasa and Beauchemin [10] have shown that the combination of HT and condensed tannin at 1.5% dietary DM also tended to reduce CH4 emissions without adverse effects on performance. Sadarman et al. [192] found that adding tannin to both acacia and chestnuts had no significant influence on DM, OM degradation, total volatile fatty acids (tVFA), NH3, and pH. The accumulation of gas at 8, 12, and 24 h was influenced by acacia tannin. They concluded that adding tannins up to levels 2%, acacia, and chestnuts did not interfere with the fermentation of rumen and in vitro by-products of soy sauce. Toral et al. [193] observed that nutrition tannins boosted milk content by numerous FA ramification isomers of 18/1 and 18/2, but that they were not effective in changing milk FA composition in the long run, especially when combined with quebracho tannins in a linoleic acid-rich diet (Figure 6).

### 4.4. Effect of Tannins on Milk Production and Composition

While the daily milk yield was 16.5 kg in dairy cows fed *L. corniculatus* with a tannin content of 27 g/kg DM, a yield of 13.8 kg in animals fed *L. corniculatus* and polyethylen glycol (PEG) was found [194]. Dschaak, et al. [195] reported that cows fed a ration supplemented with condensed tannin extract (CTE) had a decrease in milk urea nitrogen and ruminal ammonia nitrogen concentrations, without any loss in milk protein yield. Dairy cows fed *L. corniculatus* produced milk 60 percent greater than perennial ryegrass-fed cows, with a 10 percent increase in milk protein. PEG responses showed that the action of condensed tannin could explain half of the lotus effect [78].

Condensed tannin of lactating *L. corniculatus* ewes had little effect on early lactate milk production but enhanced whole milk, lactose, and protein secretion rates by 21, 12, and 14 percent, respectively, during mid to late lactation [143]. Studies assumed that the increase in milk production is due to condensed tannin content, while the content does not affect the increased voluntary feed intake (VFI). In these studies, it is clearly seen that tannin has a positive effect on milk production. However, more studies are needed to determine at what levels tannin has a positive effect in the diet. It has been accepted that the nutritional value of milk fat in ruminants can be improved by manipulating rumen biohydrogenation by changing diet-related factors. In recent years, most of the research for modulators of ruminal biohydrogenation focused to secondary plant compounds such as tannins. Both in vitro and in vivo study results show that both CT and HT can affect rumen biohydrogenation. Most of them, or their metabolites, can be found in milk and cheese [196]. However, these results are also controversial. Some in vitro studies have reported accumulation of VA and a decrease in stearic acid (18:0, SA) in rumen fluid and rumen bacteria after incubations with tannin sources such as CT or HT at the ruminal biohydrogenation stage [197,198,199,200].

Sulla is one of the most studied tannin sources in terms of its effect on the FA profile of ruminant meat and milk. Addis, et al. [201] reported that milk from sulla-fed sheep had higher levels of short-chain FA and LNA and higher levels of short-chain FA and LNA than milk from sheep fed Lolium rididum Gaudin, Medicago polymorpha L. and Chrysanthemum coronarium L., it was observed that it showed an atherogenicity index, but the 18:1 cis-9 content was lower. It was observed that dietary supplementation with quebracho (Schinopsis balansae) tannin extract (150 g/day, 0.45% DM intake) had no effect on milk FA composition of dairy cows [202]. Studies on the effect of tannins on the milk FA profile in ruminants show a wide variation in ration sources and inclusion levels of tannins, supplementation time, ration composition and physiological stages of the animal, resulting in controversial results. Although tannins are heterogeneous compounds, their structures and sizes are variable. Moreover, their metabolism and activity would be expected to depend on the tannin type. This helps to explain the controversial results regarding the ability of ruminant-derived products to modulate the FA profile.

### 4.5. Effect of Tannins on Meat Production and Composition

Most of the research reported that the intramuscular fat content was not affected by the dietary treatment explaining this lack of differences with the diet formulation. In fact, all experimental diets that compared a control group to an experimental group with addition of tannins had isonitrogenous and isoenergetic diets. Differently, the intramuscular fatty acid composition was clearly affected by the inclusion of tannins in the diet in most of them.

In ruminants, muscle fatty acids deposition depend both on the intake of different fatty acids and on the extent of the ruminal biohydrogenation of the ingested PUFA [203]. Fatty acids found in ruminant meat can be derived from two main ways, direct uptake of preformed fatty acids then transported in plasma and a de novo synthesis directly in body tissue [204].

The preformed fatty acids derived directly by digestive tract absorption, so they have a dietary or ruminal origin [205]. Differently, there is a low contribution, if present, of body fat mobilization [206]. Once in the rumen, fat is hydrolyzed by lipases and free fatty acids are released [207]. After this, biohydrogenation by rumen microorganism occurred to saturate fatty acids and reduce their toxicity (Figure 7) [205]

However, it must be considered that the biohydrogenation process resulted incomplete and some intermediate metabolites can leave the rumen and be absorbed in the intestinal tract and then incorporated in meat, with relevant consequences for consumers [208]. This aspect is the most important for nutritionists because the possibility to modulate the biohydrogenation represents an opportunity to decrease the percentage of disappearance of the dietary PUFA Moreover, some rumen microorganisms have specific synthetases for fatty acids able to synthesize odd- and branched-chain fatty acids that we can find in meat [209].

The adipose tissue is a great active site of fatty acids de novo synthesis. The main substrates derived from carbohydrates rumen fermentation and are represented by acetate and lactate [204] and they give origin to the formation of fatty acids with up to 16 carbons, with the predominant production of palmitic acid (16:0) [210], that can be after desaturated or elongated [211]. The stearoyl-CoA desaturase represented the most active desaturase [212]. A great proportion of 16:0 is converted to stearic acid (18:0) [210]. The 18:0 can be than desaturated to *cis*-9 18:1 (oleic acid) [212] in palmitoleic acid (*trans*-9 16:1) and to *cis*-9 *trans*-11 CLA. [213]

It is evident that tannins act at rumen level modulating the lipid metabolism and affecting the meat fatty acid profile [198,214]. This could probably be due to changes that tannins can induce in microbial rumen population, although it is not well known which are really involved in which specific microorganism are involved in ruminal biohydrogenation of dietary PUFA [215,216]. Different studies have tried to examine tannins effect on fatty acids deposition in meat, but different results have been observed probably due to diversity of tannins, concentration, species, and time of diet assumption. However, is evident a reduction in the biohydrogenation extent of dietary PUFA, with most of the results reporting an increase in them. Most studies reported increases in 18:2n-6 and 18:3n-3 concentrations in meat [8,200]. However, low levels of tannin addition in diet can also lack in affecting dietary PUFA proportions in meat [217,218,219,220,221]. However, the mechanisms explaining the abovementioned effects remain uncertain, although it seems to be assumed that the rumen microbiota modulation represents the best way to mediate meat fatty acid profile [222].

Tannins can also modify the concentration of some biohydrogenation intermediates. In fact, tannins can inhibit the saturation of vaccenic acid [223], although the capacity to increase *trans*-11 18:1 and *cis*-9 *trans*-11 CLA concentration is still controversial. In some studies, vaccenic acid was increased with no effect on *cis*-9 *trans*-11 CLA [200,221], while other studies reported increased *cis*-9 *trans*-11 CLA with no effect on vaccenic acid [223,224].

It must be considered that lipid diet level result important to have magnification of tannins effect on biohydrogenation. With low lipid diet, the most used in experimental trials, there is poor biohydrogenation substrate that can put in evidence the tannins effect on the process. The effects of tannins on de novo synthesized fatty acids are not well studied; nevertheless it is reported that tannins can be able to mitigate the inhibition of de novo fatty acids synthesis in meat [225].

### 4.6. Effect of Tannins on Wool Production

Access to sulfur-containing amino acids has a very effective role in wool production. Clean wool is mostly protein with a high cysteine concentration. Studies demonstrated that activation of condensed tannin in *L. pedunculatus* and *L. corniculatus* enhanced the rate of irreversible cystine loss from blood plasma, owing to lower sulfur-containing amino acids breakdown in the rumen [226,227].

Wool growth responses to condensed tannin impact vary on both the concentration and type of CT, with *L. corniculatus* growth exceeding 10% in the 22-38 g CT/kg DM range. When the condensed tannin level rose over 50 g/kg DM, the reactions seemed to be negative effects, particularly for *L. pedunculatus* and sulla. However, when condensed tannin levels were below 22 g CT/kg DM, the response to wool growth varied. Therefore, positive CT-effects for wool production in *L. corniculatus* occur between 22 and 38 g CT/kg DM [78].

### 4.7. Effect of Tannins on Reproduction

One of the important factors affecting reproduction in sheep is nutrition. Ovulation rates in sheep grazing *L. corniculatus* was found to be 22% higher on average than those grazing meadow grass and clover. This probably was due to condensed tannin content of *L. corniculatus*, determined with PEG, that is able to increase the lambing rate due to the increased fertilized rate of oocytes [78].

Downing and Scaramuzzi [228] found that the ovulation rate (OR) in sheep is influenced by nutritional levels, although the mechanism(s) through which nutrition affects OR remain unclear. The OR depends on the animal nutrition that probably relates to the ovary’s metabolic, hormonal control [228]. Changes in the plasma concentrations of growth hormone, insulin, and insulin-like are compatible with changes in biosphere energy and protein balance and muscle protein synthesis produced by nutrition. These factors can directly or by modulating ovarian gonadotropin function affect ovarian function [229]. More studies have shown that increases in ovulation rate are associated with the branched-chain plasma amino acid (BCAA) and essential amino acids (EAA) levels [143]. At the same time, intravenous infusion branched-chain plasma amino acid was increased. These findings are supported by those of Al-Gubory et al. [230], who found that the BCAAT cytosolic enzyme may directly stimulate the ovaries, raising ovulation rate through an unknown mechanism, as a result, it reduces the growth of specific BCAAA activity in the skeletal muscles of sheep [229]. BCAA is also highly sensitive to insulin in muscle protein [231]. Furthermore, only the ovaries found BCAA transferase (BCAAT) iso-enzyme [232]. These findings are confirmed by the results from Al-Gubory, Garrel, Faure and Sugino [230], which show that the developmental reduction in specific BCAAA activity in the skeletal muscles of the sheep is due to the BCAAT cytosolic enzyme, which can directly stimulate the ovaries, increasing the ovulation rate by an unknown mechanism.

### 4.8. Effect of Tannin on Parasites in the Digestive System

Parasites in the digestive system of grazing animals are one of the factors that cause significant losses in ruminants. In animals contaminated with parasites, reductions in growth, milk yield, fleece yield and reproduction occur [233]. Anthelmintic drugs are generally used to control parasites in the digestive system. The presence of these drug residues in animal products makes consumers think [234]. Therefore, the inclusion of plant species that reduce the number and effects of parasites in the diet reduces the use of these drugs, e.g.; it is known that lantana camara (Sage tea) is an important herb in the control of parasites and nematodes in the digestive system. Eucalyptus species are reported to have anthelmintic effects in goats [235].

It has been reported that condensed tannins exert their antiparasitic effects by directly inhibiting the larval development of digestive tract parasites and indirectly by binding to proteins in the rumen and preventing microbial degradation. As a result, it is thought that they improve the immunity of the host animal by providing the passage of amino acids into the duodenum and increasing protein digestion. The consumption of rations containing tannin in lambs and sheep infected with parasites increased the live weight gain and caused a decrease in the number of parasite eggs laid with manure [78]. The number of eggs in the manure obtained from sheep consuming feeds containing condensed tannins decreased by approximately 20–50% [78]. Condensed tannin is thought to either act as an anthelmintic in these plants or cause a suitable environment for microorganisms. More studies are needed to determine the amount of tannin that should be present in the diet to take advantage of the negative effects of tannins on parasites.

Plants rich in condensed tannins with demonstrated activity against gastrointestinal nematodes in sheep and goats include *Lespedeza cuneata* (sericea lespedeza), *Onobrychis viciifolia* (sainfoin), *Hedysarum coronarium* (sulla), *Lotus pedunculatus* (big trefoil), and *Lotus corniculatus* (birdsfoot trefoil). These plants can be used to reduce fecundity of nematodes but did not offer a control of them [236].

## 5. Conclusions

Tannin is a phenolic compound that tends to form compounds with nutrients found in feeds. Tannin, which is found in various proportions in plants, has negative and positive effects on animals.

Condensed tannin and feed sources containing tannin can be added to the ration at certain rates in order to enable ruminants to make better use of proteins and to reduce losses caused by excessive fragmentation of proteins in the rumen. This ratio should not negatively affect microbial protein synthesis in the rumen. In addition, the amount of tannin added to the ration should be at a level that will not have a toxic effect on animals. The determination of the tannin content of the feeds to be added to the ration has an important place in determining the amount of tannin to be added. In order to reduce the negative effects of tannin on animals and rumen microorganisms, compounds such as PEG can be added to the diets containing high levels of tannin.

Regarding tannin type and concentration, its effect can differ in animal performance. In contrast, hydrolyzable tannin can efficiently increase bypass protein, resulting in high performance in ruminants such as milk yield and milk production. Additionally, tannin can be controlled by bloat problems and intestinal parasites in grazing ruminants. However, more access to information about the effects of tannin in animal nutrition requires further investigation.

## Figures and Tables

**Figure 1 molecules-27-08273-f001:**
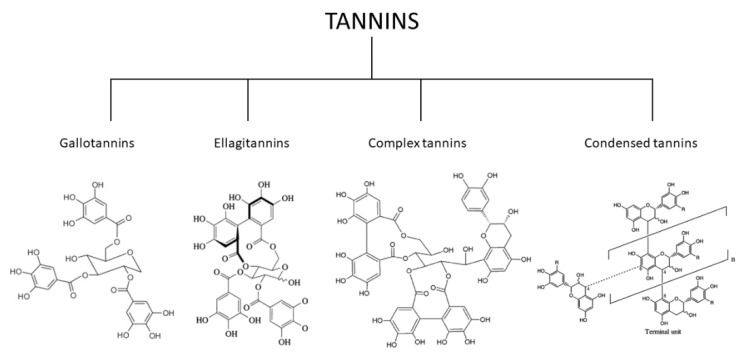
Classification of the tannins [96,97].

**Figure 2 molecules-27-08273-f002:**
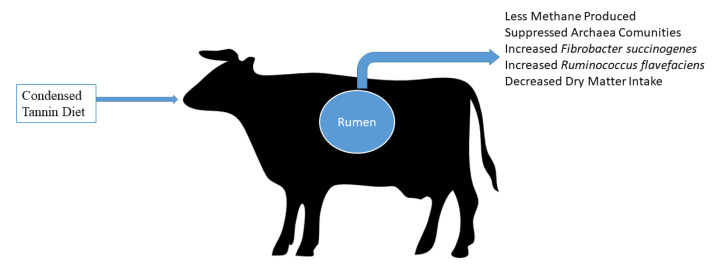
Main effects of dietary inclusion of condensed tannins.

**Figure 3 molecules-27-08273-f003:**
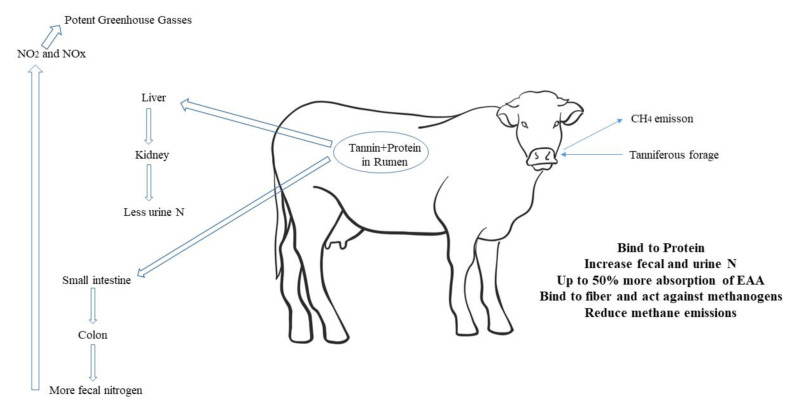
Main effect of tannins binding proteins in rumen.

**Figure 4 molecules-27-08273-f004:**
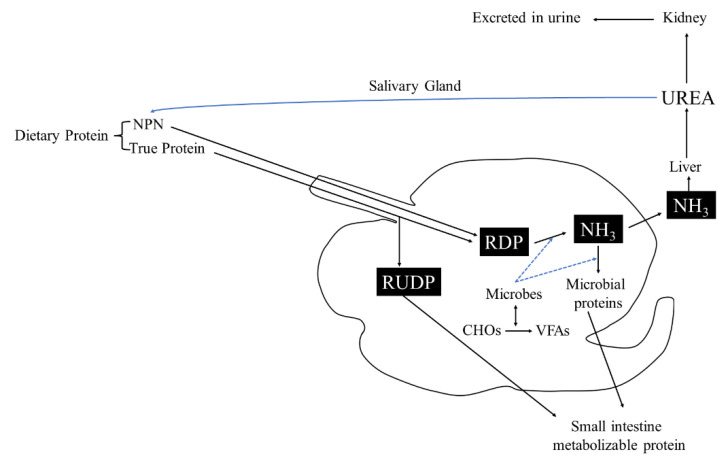
Urea cycle in rumen. RUDP: rumen undegradable protein; RDP: rumen degradable protein.

**Figure 5 molecules-27-08273-f005:**
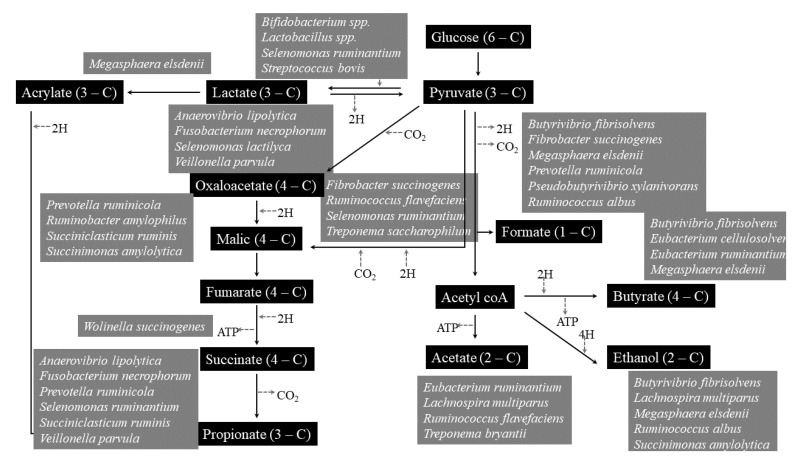
Main microbes involved in rumen fermentation processes.

**Figure 6 molecules-27-08273-f006:**
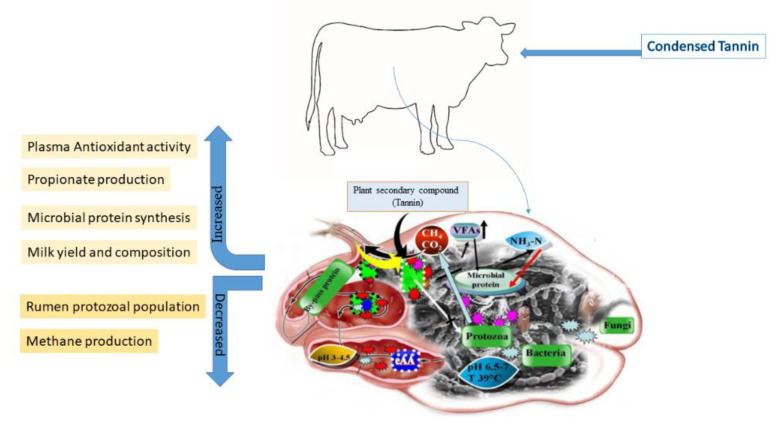
Main positive effect of plant secondary compounds assumed by diet.

**Figure 7 molecules-27-08273-f007:**
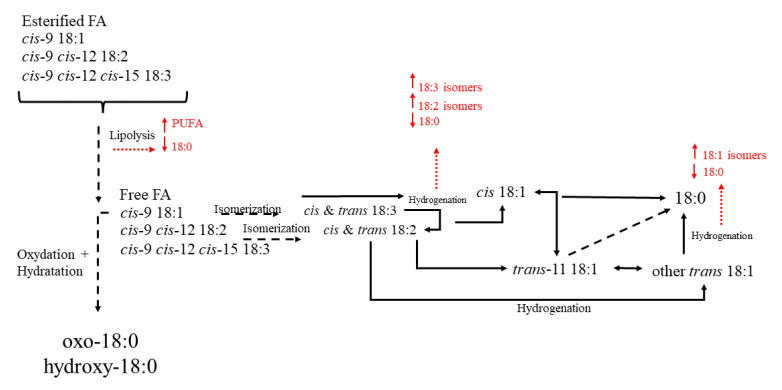
Main biohydrogenation processes in the rumen.

**Table 1 molecules-27-08273-t001:** Potential tannins’ source for animal feeding reported with their scientific name, common name, part of the plant richest in tannins and tannins’ chemical profile. Traditional and ancient vegetable tannins source used for ruminants [12,13,14,15].

Scientific Name	Common Name	Part of the Plant	Main Tannins
*Acacia mearnsii*	mimosa, wattle	barks	condensed
*Betula spp*	birch	barks	condensed
*Caesalpinia coriaria*	divi-divi	Pods	hydrolizable
*Castanea sativa*	chestnut, sweet chestnut	wood	hydrolizable
*Coriaria myrtifolia*	mediterrenean coriaria (emborrachacabras, redoul, roldor, rodor)	leaves	condensed/hydrolizable
*Cotinus coggygria (syn Rhus cotinus)*	smoke tree	leaves	hydrolizable
*Larix*	larch	barks	condensed
*Mirtus communis*	myrtle	leaves	hydrolizable
*Picea abies*	norway spruce	barks	condensed
*Pinus halepensis*	Aleppo pine	barks	condensed
*Quercus aegilops*	valonea oak, Turkish oak	acorn cups	hydrolizable
*Quercus coccifera*	garouille	husk of root	hydrolizable
*Quercus infectoria*	Aleppo oak	galls	hydrolizable
*Quercus ilex*	holm oak	barks	condensed/hydrolizable
*Quercus spp. (Q. ilex, Q. robur, Q. petraea, Q. pyrenaica)*	oak	barks	condensed/hydrolizable
*Quercus suber*	cork oak	inner bark	condensed/hydrolizable
*Rhus coriaria*	sumac	leaves	hydrolizable
*Salix spp.*	willow	narks	condensed
*Schinopsis balansae, S. lorentzii*	quebracho	wood	condensed
*Terminalia chebula*	myrabolans	fruits	hydrolizable

**Table 2 molecules-27-08273-t002:** Potential tannins’ source for animal feeding reported with their scientific name, common name, part of the plant richest in tannins and tannins’ chemical profile. Fodder and plants tannins source used for ruminants [15,16,17,18,19,20,21,22,23,24,25,26,27,28,29,30,31,32,33,34,35,36,37,38,39,40,41,42,43,44,45,46,47,48,49,50].

Scientific Name	Common Name	Part of the Plant	Main Tannins
*A. mearnsii*	black wattle	leaves	condensed
*A. nilotica*	gum Arabic tree	leaves	condensed
*C. sativa*	hemp	leaves/flowers	hydrolizable
*J. regia*	common walnut	leaves/flowers	hydrolizable
*L. corniculatus*	bird’s-foot trefoil	leaves/flowers	condensed
*L. pedunculatus*	marsh bird’s-foot trefoil	leaves/flowers	condensed
*P. abies*	European spruce	leaves	condensed
*P. granatum*	pomegranate	fruit	hydrolizable
*Q. robur*	European oak	leaves	hydrolizable
*R. coriaria*	tanner’s sumach	leaves	condensed
*R. fruticosus*	European blackberry	fruit	hydrolizable
*S. lorentzii*	red quebracho	fruit	condensed/hydrolizable
*S. balansae*	willow-leaf red quebracho	fruit	condensed
*T. chebula*	black- or chebulic myrobalan	fruit	hydrolizable
*V. vinifera*	common grape vine	fruit	condensed

**Table 3 molecules-27-08273-t003:** Potential tannins’ source for animal feeding reported with their scientific name, common name, part of the plant richest in tannins and tannins’ chemical profile. Vegetal industry co-products tannins source used for ruminants [49,51,52,53,54,55,56,57,58,59,60,61,62,63,64,65,66,67,68,69,70,71,72].

Scientific Name	Common Name	By-Product	Main Tannins
*Cupressus lusitanica*	Mexican cedar	steam distillation residues	condensed/hydrolizable
*Cistus ladanife*	labdanum	steam distillation residues	condensed/hydrolizable
*Coffea arabica*	coffee	pulp	condensed
*P. Granatum*	pomegranate	peels	condensed/hydrolizable
*Vitis vinifera*	red grape variety	pomace	condensed
*Castanea sativa*	chestnut	shells	condensed/hydrolizable
*Camellia sinensis L.*	tea	leaves	condensed/hydrolizable
*Myrtus communis*	common myrtle	leaves	condensed/hydrolizable
*Endopleura uchi*	yellow uxi	bark	condensed/hydrolizable
*Picea abies*	Norway spruce	bark	condensed
*Picea abies*	spruce	bark	condensed/hydrolizable
*Eucalyptus globulus*	blue gum	leaves	condensed/hydrolizable
*Pinus taeda*	loblolly pine	bark	condensed/hydrolizable
*Persea americana*	avocado	peel/pulp	condensed
*Musa acuminata*	banana	peel/seed/pulp	condensed
*Psidium guajava*	guava	peel/seed/pulp	condensed
*heterophyllus artocarpus*	jackfruit	peel/seed/pulp	condensed
*Dimocarpus longan*	longan	peel/seed/pulp	condensed
*Mangifera indica*	mango	peel/seed	condensed
*Olra europae*	olive	leaves	hydrolizable
*Cynara cardunculus*	artichoke	leaves	condensed
*Citrus limon*	lemon	pomace	condensed
*Brassica napus*	canola	pulp	condensed

**Table 4 molecules-27-08273-t004:** Statistical significance of perceived nutritional digestibility and ruminal states in sheep given dry grape by-products-based diets [144].

Nutrient	Trt ^1^	Contrasts (*p*<)		
		Linear	Quadratic	Cubic
Dry Matter	0.09	0.69	<0.05	<0.05
Organic Matter	0.65	0.32	0.53	0.44
Crude protein	<0.05	<0.05	0.96	0.72
Natural detergent fiber	<0.01	<0.05	0.83	0.60
Acid detergent fiber	<0.01	<0.05	0.52	0.58

^1^ Significant level of treatment effect.

## Data Availability

Not applicable.

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
