# Peer review of "Tannin in Ruminant Nutrition: Review"

_molecules, 2022, doi:10.3390/molecules27238273_

Round 1

Reviewer 1 Report

Manuscript ID: molecules-1988106

Title: Tannin in Ruminant Nutrition: Review

This review article summarizes the use and effects of tannins in ruminant nutrition. In general, the authors should add some elements that make the manuscript more fluent for the reader (tables, figures); the risk is to bore the reader. 

Additional work needs to be done in order to prepare this review for publication.

My detailed comments are presented below.

The total number of Figures and Tables (only two) is very low for this type of manuscript. Please think about at least one-two more to 1) catch the reader’s attention 2) stress the most important aspects. For example, you can add a figure illustrating the main sources of tannins…

In my opinion, you could also add a table summarizing various previous studies conducted in tannins and their effect in ruminants.

Figure 1: first of all, please improve its quality. In my opinion, the description of the four categories should be presented as a footnote under the table, and not in the main text.

Line 96: correct in “gallotannins”

Paragraph number 3 “Chemical Properties of Tannins” (Lines 87-144): to make the reading more fluent, you could add a figure in which you schematize the chemical reactions.

Lines 109 and 116: delete the names in literature references by adding numbers

Line 165: for “Sericea lespedeza” please use italic

Paragraph 4 “Effect of Tannins on Ruminant Nutrition”: first of all, in my opinion this paragraph should be extended by adding other references. Secondly, the other paragraphs (from 5 to 11) should be transformed into sub-paragraph of the main paragraph (n. 4). Use the number 4.1, 4.2, etc.

Line 194: correct “type” in “species”

Line 197: use the acronym HT for “Hydrolyzable tannins” the first time you cite this term in the manuscript. Check throughout the manuscript.

Lines 236 and 340: Cite as “Cabral Filho et al.” [57]

Lines 238-243: how the authors explain these results?

Line 255: please explain the acronym CP the first time you use it (also for “DM”, “OM”, etc)

Line 276: cite as “Hagerman et al.”

Line 280: Please shift “PEG” near polyethanol-glycol

Line 322: please delete the name in reference and add number

Line 348: use “2%”

Lines 368-370: please rewrite the sentence

Line 369 and 393: for “L. corniculatus” please use italic. Please check throughout the manuscript and correct all terms non written in italic

Line 379: explain the acronym “EAA”

Line 381: add the number of reference (Al-Gubory et al. 2012)

Line 391 - Paragraph 10 (“Milk production”): since you presented some results concerning the effect of tannins in milk production, I think that it would be interesting add some results about the other products produced by ruminants (meat).

Paragraph 11, lines 431-451: the authors could cite some sources of tannins used for this aim (anthelimintic effect)

Author Response

Dear reviewer,

thank you for your precious work. Below our reply to your comments

Reviewer 1

This review article summarizes the use and effects of tannins in ruminant nutrition. In general, the authors should add some elements that make the manuscript more fluent for the reader (tables, figures); the risk is to bore the reader. 

Additional work needs to be done in order to prepare this review for publication.

My detailed comments are presented below.

The total number of Figures and Tables (only two) is very low for this type of manuscript. Please think about at least one-two more to 1) catch the reader’s attention 2) stress the most important aspects. For example, you can add a figure illustrating the main sources of tannins…

In my opinion, you could also add a table summarizing various previous studies conducted in tannins and their effect in ruminants.

 AU: Dear reviewer, thank you for your suggestion, More figures and tables are now present in the review.

Figure 1: first of all, please improve its quality. In my opinion, the description of the four categories should be presented as a footnote under the table, and not in the main text.

AU: the figure was revised and upload of higher quality. Thank you for your suggestion

Line 96: correct in “gallotannins”

AU: Done

Paragraph number 3 “Chemical Properties of Tannins” (Lines 87-144): to make the reading more fluent, you could add a figure in which you schematize the chemical reactions.

AU: We added a lot of figures but bot this one because is not the main topic of the revies. However ew tried to be more explicative with figures added

Lines 109 and 116: delete the names in literature references by adding numbers

AU: Done

Line 165: for “Sericea lespedeza” please use italic

AU: Done

Paragraph 4 “Effect of Tannins on Ruminant Nutrition”: first of all, in my opinion this paragraph should be extended by adding other references. Secondly, the other paragraphs (from 5 to 11) should be transformed into sub-paragraph of the main paragraph (n. 4). Use the number 4.1, 4.2, etc.

 AU: Done

Line 194: correct “type” in “species”

AU: Done

Line 197: use the acronym HT for “Hydrolyzable tannins” the first time you cite this term in the manuscript. Check throughout the manuscript.

AU: Done

Lines 236 and 340: Cite as “Cabral Filho et al.” [57]

AU: Done

Line 255: please explain the acronym CP the first time you use it (also for “DM”, “OM”, etc)

AU: Done

Line 276: cite as “Hagerman et al.”

AU: Done

Line 280: Please shift “PEG” near polyethanol-glycol

AU: Done

Line 322: please delete the name in reference and add number

AU: Done

Line 348: use “2%”

AU: Done

Lines 368-370: please rewrite the sentence

AU: Done

Line 369 and 393: for “L. corniculatus” please use italic. Please check throughout the manuscript and correct all terms non written in italic

AU: Done

Line 379: explain the acronym “EAA”

AU: Done

Line 381: add the number of reference (Al-Gubory et al. 2012)

 AU: Done

Line 391 - Paragraph 10 (“Milk production”): since you presented some results concerning the effect of tannins in milk production, I think that it would be interesting add some results about the other products produced by ruminants (meat).

 AU: Done

Paragraph 11, lines 431-451: the authors could cite some sources of tannins used for this aim (anthelimintic effect)

AU: Done

Reviewer 2 Report

The paper was well prepared and covered most relevant aspects, only small issues require addition as follows,

1. Section of Effect of tannins on average daily gain, growth rate and carcass quality in ruminants.

2. Effect of tannins on rumen microbiomes, fermentation and methane production..

The above topics need to be added and/or focussed.

Author Response

DEar reviewer,

thank you very much for your work. Below our reply to your comments

Reviewer 2

The paper was well prepared and covered most relevant aspects, only small issues require addition as follows,

  1. Section of Effect of tannins on average daily gain, growth rate and carcass quality in ruminants.

AU: we decided to not write this section because in most of the literature the short time application and/or the dose and/or the diet can’t justify differences due to tannins. However, thank you for this suggestion.

  1. Effect of tannins on rumen microbiomes, fermentation and methane production..

AU: Thank you for this suggestion, we add section on this topic.

The above topics need to be added and/or focussed.

Round 2

Reviewer 1 Report

The authors have improved their work. 

I think the manuscript could be accepted in the present form for publication.

Author Response

Thank you for your work, it has been appreciated